# Exploring the Potential of Voxel-Mirrored Homotopic Connectivity (VMHC) and Regional Homogeneity (ReHo) in Understanding Cognitive Changes After Heart Transplantation

**DOI:** 10.3390/biomedicines13040873

**Published:** 2025-04-03

**Authors:** Qian Qin, Jia Liu, Wenliang Fan, Xinli Zhang, Jue Lu, Xiaotong Guo, Ziqiao Lei, Jing Wang

**Affiliations:** 1Department of Radiology, Union Hospital, Tongji Medical College, Huazhong University of Science and Technology, Jiefang Avenue #1277, Wuhan 430022, China; qinqian2022@163.com (Q.Q.); liujia_1990512@foxmail.com (J.L.); fwl@hust.edu.cn (W.F.); z1694342748@163.com (X.Z.); xhlujue@hust.edu.cn (J.L.); m202476518@hust.edu.cn (X.G.); 2Hubei Provincial Clinical Research Center for Precision Radiology & Interventional Medicine, Wuhan 430022, China; 3Hubei Key Laboratory of Molecular Imaging, Wuhan 430022, China

**Keywords:** heart transplantation, cognitive impairment, resting-state functional magnetic resonance imaging, voxel-mirror homotopic connectivity, regional homogeneity

## Abstract

**Objective**: This study aimed to investigate the application value of voxel-mirrored homotopic connectivity (VMHC) and regional homogeneity (ReHo) in evaluating cognitive impairment after heart transplantation. **Methods**: A total of 68 heart transplant patients and 56 healthy controls were included. ReHo and VMHC were calculated using DPARSF software. A two-sample *t*-test was applied to compare the differences in ReHo and VMHC between the two groups, and a Pearson correlation analysis was performed by extracting the VMHC and ReHo values of different brain regions and correlating them with cognitive scale scores of the patient groups. **Results**: Mini-Mental State Examination (MMSE) and Montreal Cognitive Assessment (MoCA) scores were lower in the heart transplant group than in the control group (MMSE: t = 4.028, *p* < 0.001; MoCA: t = 4.914, *p* < 0.001). Compared with the control group, the ReHo values of Frontal_Sup_R (t = −4.422, *p* < 0.001), Thalamus_L (t = −3.911, *p* < 0.001), and Calcarine_L (t = −3.640, *p* < 0.001) were lower in the heart transplantation group, while the ReHo of Temporal_Sup_L was higher (t = 4.609, *p* < 0.001). VMHC was elevated for bilateral Cerebellum_Crus1 (t = 3.803, *p* < 0.001) and decreased for bilateral calcarine (t = −3.424, *p* < 0.001). The ReHo of Frontal_Sup_R was positively correlated with MMSE (*r* = 0.345, *p* = 0.004) and MoCA (*r* = 0.376, *p* = 0.002). The ReHo of Temporal_Sup_L was also positively correlated with MMSE (*r* = 0.397, *p* < 0.001) and MoCA (*r* = 0.542, *p* < 0.001). The VMHC of bilateral calcarine showed a positive correlation with MMSE (*r* = 0.513, *p* < 0.001) and MoCA (*r* = 0.398, *p* < 0.001). Other differential brain regions showed no significant correlation with the MMSE and MoCA scale scores. **Conclusions**: Cognitive decline was observed in heart transplant patients. Heart transplant patients exhibited altered ReHo and VMHC in several brain regions compared with healthy controls. These changes may underlie impaired cognitive function in heart transplant patients. These findings may contribute to understanding the neural mechanisms of cognitive changes in heart transplant patients and could inform future research on potential intervention strategies.

## 1. Introduction

Heart transplantation is a surgical procedure primarily for advanced congestive heart failure and severe coronary artery disease and is the primary treatment for end-stage heart disease [1]. According to statistics, the 1-year survival rate after heart transplantation is 87%, the 5-year survival rate is 72%, and the 10-year survival rate is expected to be 55% to 60% [2]. While heart transplantation improves overall patient function and quality of life, patients still face many challenges. Postoperative cognitive dysfunction (POCD) is a common postoperative reversible acute psychiatric disorder in heart transplant patients [3]. POCD refers to the state of cognitive impairment that develops in patients after surgical procedures, and the incidence of this condition varies (15% to 50%) depending on the risk and type of surgery [4]. The clinical features of the patients include attention deficits, impaired consciousness, changes in cognitive abilities, and disruption of sleep–wake cycles, which seriously affect the quality of life of the patients after surgery [5]. The occurrence of POCD in heart transplant patients not only increases the patient’s medical costs and prolongs the patient’s hospitalization, but also increases the patient’s mortality [6]. Therefore, in recent years, organ transplant physicians have become increasingly concerned about the occurrence of POCD after organ transplantation. Up to 40% of heart transplant patients develop cognitive impairment [7]. However, the neural mechanisms underlying cognitive impairment in heart transplant patients remain unclear.

In recent years, resting-state functional MRI (rs-fMRI) has been widely used in the study of the functional connectivity of the resting-state human brain [8]. Studies have shown that some brain regions or brain networks play an important role in human cognitive functions, such as the default mode network (DMN), which is not only involved in primary cognitive functions such as spatial location, sensation, and attention in the cognitive process [9], but is also closely related to higher cognitive functions such as social cognition, semantic memory, and logical thinking [10]. The application of some indicators in rs-fMRI to evaluate the differences in brain area activation and functional connectivity in patients with cognitive impairment is a hot research topic at this stage. Among them, VMHC is a method used to analyze the homotopic connectivity between a given voxel in the cerebral hemisphere and its mirror voxel in the contralateral hemisphere, which can reflect the functional connectivity of the brain in the resting state and is used to assess the endogenous spontaneous activity of neurons with the same origins in both cerebral hemispheres [11]. ReHo can reflect the time-series similarity between each voxel and its neighboring voxels [12]. The above metrics are effective methods for assessing functional connectivity and spontaneous activity in brain regions, but their application in evaluating impaired cognitive function in heart transplantation patients still needs to be further investigated and expanded. Therefore, the aim of this study is to apply VMHC and ReHo and combine them with cognition-related scales to provide an imaging basis for the pathogenesis of cognitive impairment in heart transplant patients.

## 2. Materials and Methods

### 2.1. Objects

Patients who underwent heart transplantation at Union Hospital of Huazhong University of Science and Technology from September 2021 to September 2023 were selected for the study. Inclusion criteria were (1) habitual right-handedness, (2) ability to communicate and understand questionnaires, (3) successful heart transplantation with an expected survival period of more than one month after discharge from the hospital, and (4) no preoperative cognitive impairment. Exclusion criteria included (1) a history of substance dependence, (2) organic brain diseases such as traumatic brain injury, epilepsy, or severe physical diseases, (3) contraindication to magnetic resonance examination (MRI), (4) joint organ transplantation, and (5) suffering from hereditary or congenital diseases. Community health volunteers were recruited as the control group during the same period. Inclusion criteria for the controls were (1) Han nationality, (2) right-handedness, and (3) no family history of mental illness. Exclusion criteria for the controls were contraindications to magnetic resonance examination. The enrollment was assisted by a cardiac surgeon with 8 years of experience, who collected basic information of all enrolled patients, such as gender, age, and years of education. All organs were donated voluntarily after death, in accordance with the Chinese Regulations on Human Organ Transplantation. No organs from death row inmates were used. All subjects or their guardians signed an informed consent form. A total of 68 patients and 57 controls were initially included. The study adhered to the Declaration of Helsinki and was approved by the Hospital Ethics Committee (approval number: 2022(0166-01)).

### 2.2. Cognitive Assessment

Cognitive function was assessed in all enrolled patients using the Mini-Mental State Examination (MMSE) and the Montreal Cognitive Assessment (MoCA). The MMSE measures general cognitive function and comprises six domains: orientation, registration, attention and calculation, recall, language, and visuospatial functioning. The maximum score for each domain ranges from 1 to 10, with a total possible score of 30, where higher scores indicate better general cognitive functioning [13]. The MoCA is a brief test of cognitive functioning and comprises seven domains: short-term memory, visuospatial functioning, executive functioning, attention, concentration, working memory, language, and orientation [14]. The test takes approximately ten minutes, with a total possible score of 30, where higher scores indicate better overall cognitive functioning.

### 2.3. MRI Data Acquisition

MRI data, including anatomical and functional images, were acquired using a Siemens Trio Tim 3.0T MRI system and a 32-channel head coil. Subjects were instructed to close their eyes, remain awake and relaxed, and wear headphones to minimize scanning noise. Foam pads were used to minimize head movement and prevent motion artifacts. Anatomical images were obtained using a high-resolution T1-weighted MP-RAGE sequence with 192 sagittal slices covering the whole brain (TR = 1900 ms; TE = 2.3 ms; TI = 900 ms; flip angle = 9°; voxel size = 0.3 × 0.3 × 0.8 mm^3^; FOV = 240 mm × 240 mm; slice thickness = 0.8 mm). Functional images were acquired using a gradient-echo EPI sequence with 240 volumes (TR = 2000 ms; TE = 30 ms; flip angle = 90°; voxel size = 2.4 × 2.4 × 2.4 mm^3^; FOV = 230 mm × 230 mm; slice thickness = 2.4 mm). Two radiologists independently reviewed the MR images for abnormalities, such as hemorrhage, ischemia, and tumors. Subjects with these abnormalities were excluded.

### 2.4. Data Preprocessing

Based on Matlab 2022b, the data were preprocessed using Data Processing Assistant for Resting-State fMRI Advanced edition (DPARSFA, http://www.rfmri.org/DPARSF (accessed on 10 January 2025)) software [15] with statistical parametric maps (SPM 12, http://www.fil.ion.ucl.ac.uk/spm (accessed on 10 January 2025)). The processing steps included (1) format conversion, which converted the DICOM raw data to a NIfTI format, (2) removal of the first 10 time points of the rs-fMRI data, (3) temporal layer correction, (4) head-motion correction, which removed subjects with head movements greater than 3 mm or rotations greater than 3° (three patients were excluded in the heart transplantation group, and four patients were excluded in the healthy control group), (5) spatial normalization (the head-movement-corrected functional images were aligned to the Montreal Neuroscience Institute standard space and resampled to voxel sizes of 3 mm × 3 mm × 3 mm), (6) de-linearization drift, which removes trends in nonneural activity, (7) removal of covariates, including cerebral white matter signals, cerebrospinal fluid signals, and the 24 head-movement parameters, and (8) spatial filtering, which was applied to the data from 0.01 to 0.08 Hz.

### 2.5. Calculation of Indicators

ReHo calculation was performed using DPARSF software (Version V5.0_200401). The time-series similarity between each voxel and its neighboring 26 voxels was assessed based on Kendall’s concordance coefficient (KCC), then the ReHo metrics were smoothed with a Gaussian kernel with a half-height and width of 6 mm × 6 mm × 6 mm, and finally, the results were normalized using Fisher-Z transformation to Z-values. The VMHC was calculated using DPARSF software (Version V5.0_200401). The Pearson correlation coefficients between mirror symmetry voxels in the whole brain were calculated to obtain the VMHC, and finally, the results were normalized to Z-values using Fisher-Z transformation.

### 2.6. Statistical Analysis

The demographic data of all patients were statistically analyzed along with the scale results using SPSS 27.0. Measurement information that conformed to a normal distribution was expressed as mean ± standard deviation ($\bar{x}$ ± s), and independent samples *t*-tests were used for comparison between groups. Count data were compared between groups using χ^2^ tests. The results of rs-fMRI indexes of the two groups were evaluated using the resting-state statistical analysis model (Specify 2nd-level) of SPM 12 software. Two independent samples *t*-tests were used to evaluate the ReHo and VMHC outcome brain maps of the two groups, with head movement, age, gender, and years of education as covariates, and the results were corrected by false discovery rates (FDRs), qualifying the voxel level at *p* < 0.001 and the cluster level at *p* < 0.05. AAL-90 templates were used for the brain regions with significant differences, and the MNI coordinates of the index peak points were extracted. The mean values of ReHo and VMHC of the brain regions with differences were extracted and analyzed by Pearson correlation with MMSE and MoCA scales, with differences considered statistically significant at *p* < 0.05.

## 3. Results

### 3.1. Demographics and Scale Results

A total of 68 heart transplant patients and 56 healthy controls were included in this study. The differences in age, gender, and years of education between the two groups of subjects were not statistically significant (*p* > 0.05). The MMSE and MoCA scale scores of the heart transplant group were lower than those of the control group, and the difference was statistically significant (MMSE: t = 4.028, *p* < 0.001; MoCA: t = 4.914, *p* < 0.001). There was a significant difference between the two groups of subjects in the six MoCA domains of visuospatial/executive (t = 7.175, *p* < 0.001), naming (t = 2.519, *p* =0.014), attention (t = 3.972, *p* < 0.001), language (t = 4.090, *p* < 0.001), abstraction (t = 5.988, *p* < 0.001), and memory (t = 9.729, *p* < 0.001), as shown in Table 1.

### 3.2. ReHo Analysis Results

Compared with the control group, the heart transplantation group exhibited reduced ReHo in the Frontal_Sup_R (t = −4.422, *p* < 0.001), Thalamus_L (t = −3.911, *p* < 0.001), and Calcarine_L (t = −3.640, *p* < 0.001), while the ReHo of Temporal_Sup_L was elevated (t = 4.609, *p* < 0.001) (see Table 2 and Figure 1).

### 3.3. VMHC Analysis Results

VMHC was elevated (t = 3.803, *p* < 0.001) in the bilateral Cerebellum_Crus1 of the heart transplant group, while VMHC was reduced (t = −3.424, *p* < 0.001) in the bilateral calcarine, as shown in Table 3 and Figure 2.

### 3.4. Correlation Analysis of Differential Brain Regions with Cognitive Scales

The ReHo of Frontal_Sup_R in the heart transplant group was positively correlated with MMSE (*r* = 0.345, *p* = 0.004) and MoCA (*r* = 0.376, *p* = 0.002). The ReHo of Temporal_Sup_L was also positively correlated with MMSE (*r* = 0.397, *p* < 0.001) and MoCA (*r* = 0.542, *p* < 0.001) (see Figure 3). The VMHC of bilateral calcarine in the patient group showed a positive correlation with MMSE (*r* = 0.513, *p* < 0.001) and MoCA (*r* = 0.398, *p* < 0.001) (see Figure 4). Other differential brain regions showed no significant correlation with MMSE and MoCA scale scores.

## 4. Discussion

This study focused on the feasibility of applying rs-fMRI to evaluate cognitive impairment in heart transplant patients. We primarily utilized ReHo and VMHC analyses to assess the changes in functional connectivity and spontaneous brain activity between the cerebral hemispheres after heart transplantation and correlated these findings with MMSE and MoCA scales. The results demonstrated that the MMSE and MoCA scale scores were lower in the heart transplant group than in the control group. In the rs-fMRI results, several brain regions exhibited altered ReHo and VMHC in heart transplant patients, which were correlated with MMSE and MoCA scale scores. These findings may contribute to a better understanding of the neuropathophysiological mechanisms of cognitive dysfunction in heart transplant patients.

The MMSE and MoCA scales are the most widely used tools for evaluating cognitive function in clinical practice. In the present study, the reduced MMSE and MoCA scores in the heart transplant group indicated cognitive decline. Bucker et al. reported that approximately 40% of patients exhibited cognitive impairment at 3 years post-heart transplantation, primarily in domains such as processing speed, memory, language, and executive function [16]. Consistent with this finding, our study revealed cognitive impairment in six domains: visuospatial/executive function, naming, attention, language, abstraction, and memory. POCD is characterized by impaired memory, decreased information processing, and reduced attention, with a range of negative outcomes, including changes in mood and personality [17]. The one-year mortality rate after surgery is nearly twice as high in patients with POCD compared to those without POCD [18]. POCD is believed to involve pathological processes, including neuroinflammation, mitochondrial dysfunction, oxidative stress, blood–brain barrier (BBB) damage, neurotrophic support damage, and synaptic damage [19]. These molecular pathways are suspected to contribute to cognitive impairment after heart transplantation.

ReHo, calculated using Kendall’s concordance coefficient (KCC) to assess the consistency of adjacent voxels in the time series of rs-fMRI, indirectly reflects the synchronization of local spontaneous brain activities [20]. Previous studies have shown that patients with cognitive impairment exhibit altered spontaneous brain activity in the resting state. Zhang et al. found that ReHo in the medial prefrontal cortex and precuneus was significantly lower in patients with cognitive impairment compared to healthy subjects [21]. Liu et al. noted that ReHo in the right superior temporal gyrus and right middle temporal gyrus of patients with mild cognitive impairment (MCI) was significantly lower than that of healthy subjects [22]. In the present study, reduced ReHo in the Frontal_Sup_R, Thalamus_L, and Calcarine_L suggests disrupted local functional integration in these regions. The superior frontal gyrus is a crucial component of the prefrontal lobe, which is responsible for integrating internal and external environmental information, extracting environmental memories, and playing a central role in attention and executive function, and its decreased ReHo may directly contribute to the observed deficits in attention and abstraction tasks [23]. The thalamus acts as an information relay station for the brain and plays a key role in processing and transmitting sensory information [24]. It is also recognized as a critical node for coordinating cognitive functions. Its reduced ReHo could impair information transfer between cortical and subcortical regions, exacerbating memory and language deficits [25]. The calcarine sulcus belongs to the occipital cortex, which is involved in memory, speech, attention, and executive functions in addition to visual information processing as an important visual center [26]. This finding is generally consistent with our results, which showed that VMHC in heart transplant patients is significantly correlated with cognitive function. The resting-state brain activity in the occipital lobe is involved in the entire cognitive process beyond visuospatial structures. Conversely, we also found elevated ReHo in the Temporal_Sup_L in heart transplant patients. The superior temporal gyrus, which is part of the temporal lobe, contains the hippocampus and parahippocampal gyrus, which are associated with memory. The superior temporal gyrus also contains areas 41 and 42 and the transverse temporal gyrus, which form the auditory cortical area and are thought to be involved in cognitive processes and language function. Elevated ReHo in the superior temporal gyrus may reflect compensatory neural recruitment to mitigate language-related cognitive decline [27].

VMHC is a validated rs-fMRI evaluation method for quantifying the functional connectivity between the two cerebral hemispheres and is one of the most prominent features of the brain’s basic functional architecture. In pathological states, altered communication between the two cerebral hemispheres can significantly impact cognition and behavior [28]. In this study, VMHC was elevated in the bilateral Cerebellum_Crus1 and decreased in the bilateral calcarine in the heart transplant group. Decreased VMHC in the bilateral calcarine, a primary visual cortex, may indicate disrupted visual information processing, potentially explaining visuospatial deficits in patients. An increasing number of studies recognize that the cerebellum is not only involved in maintaining body balance, motor coordination, and eye movements [29] but also plays a significant role in sensory, cognitive, and emotional learning and regulation [30]. Elevated VMHC in the Cerebellum_Crus1 might indicate maladaptive plasticity in chronic cognitive impairment. Many studies have used fMRI to map motor and non-motor task processes and resting-state networks in the human cerebellar cortex [31]. Pagen et al. found that the functional connectivity of the cerebellum to the brain’s default mode network (DMN) was generally reduced in patients with amnestic mild cognitive impairment (MCI) [32]. The DMN is primarily responsible for the body’s cognitive control in the resting state and maintains inward thinking activity. In patients with Alzheimer’s disease, the DMN is initially disrupted by amyloid deposition caused by the disease process [33]. Both the cerebellum and the prefrontal lobe belong to the DMN. The reduced ReHo in the prefrontal lobe and elevated VMHC in the cerebellum in this study suggest that the DMN is damaged in heart transplant patients.

These alterations in ReHo and VMHC may stem from multiple pathological processes post-heart transplantation. Neuroinflammation and oxidative stress, common in post-transplant patients [19], could damage synaptic plasticity and neuronal synchronization, leading to reduced ReHo in frontal and thalamic regions. Additionally, chronic hypoperfusion due to cardiovascular instability may impair interhemispheric connectivity (VMHC), particularly in the occipital cortex. The cerebellum’s elevated VMHC might be driven by its role in compensating for cortical dysfunction through cerebellar–cortical loops [31], but prolonged overactivation could deplete neural resources, worsening cognitive fatigue.

In this study, there were significant correlations between the ReHo of Frontal_Sup_R, the ReHo of Temporal_Sup_L, and the VMHC of calcarine and the MMSE and MoCA scores in the heart transplantation group. We speculate that the pathogenesis of cognitive dysfunction in heart transplant patients may be related to these brain regions. Changes in the functional connectivity of these brain regions may be predictive of cognitive impairment in heart transplant patients. These findings have potential diagnostic and therapeutic implications. The ReHo and VMHC patterns may serve as neuroimaging biomarkers for early detection of cognitive decline in heart transplant patients. In addition, therapeutic strategies targeting specific brain regions are worth exploring such as, for example, the enhancement of thalamocortical connectivity through neuromodulation. Non-invasive techniques such as transcranial magnetic stimulation (TMS) of the prefrontal cortex have shown promise for improving executive function in other populations [34]. Monitoring changes in ReHo and VMHC in specific brain regions may be useful in assessing the effectiveness of interventions aimed at restoring cognitive control networks.

## 5. Limitations

Despite the importance of our findings, there are some limitations. Firstly, the sample size is relatively small, which may lead to biased results in differential brain regions, and the sample size will be continuously increased in subsequent studies. Secondly, the present study was cross-sectional, and it was not possible to determine the relationship between brain activity and the progression of cognitive impairment in heart transplant patients. Thirdly, most of the patients had received different types of medications before enrollment. These medications may interfere with the normal metabolism of the brain and thus have an effect on brain activity. Importantly, this study is exploratory and data-driven rather than driven by a specific, narrow research hypothesis. As such, the findings should be regarded as exploratory rather than confirmatory. The results highlight potential associations between VMHC, ReHo, and cognitive function in heart transplant patients, but they need to be replicated in larger, more diverse cohorts to validate their clinical relevance.

## 6. Conclusions

In summary, heart transplant patients exhibit significant cognitive decline and undergo a series of changes in functional brain activity. Using VMHC and ReHo can objectively and comprehensively reflect the changes in functional connectivity and spontaneous activity between the bilateral cerebral hemispheres. The positive correlation between the ReHo of Frontal_Sup_R, the ReHo of Temporal_Sup_L, and the VMHC of calcarine and cognitive function scales in heart transplant patients provides an imaging basis for further elucidating the pathogenesis of cognitive dysfunction in heart transplant patients. However, further research is needed to explore their potential as biomarkers for cognitive dysfunction in this population.

## Figures and Tables

**Figure 1 biomedicines-13-00873-f001:**
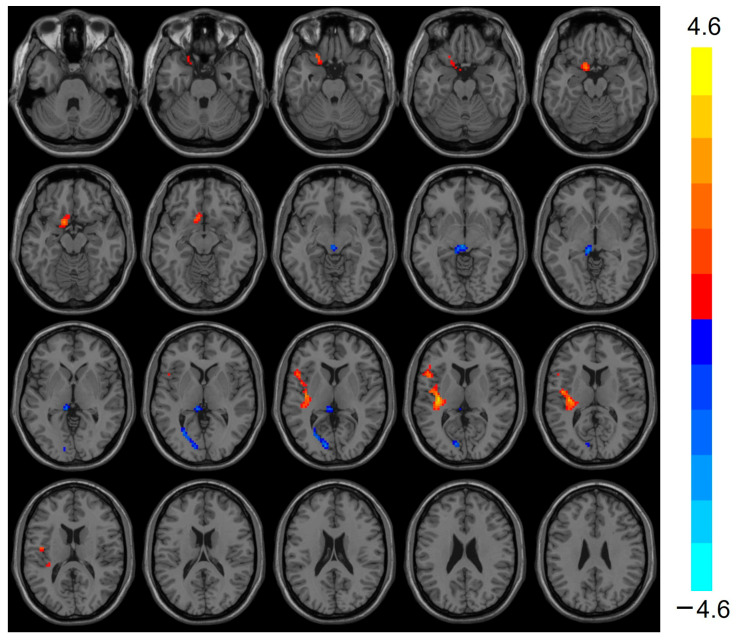
Images show decreased ReHo for Frontal_Sup_R, Thalamus_L, and Calcarine_L in the heart transplantation group (blue) and increased ReHo for Temporal_Sup_L (red). Color bars on the right show t-values. ReHo, regional homogeneity.

**Figure 2 biomedicines-13-00873-f002:**
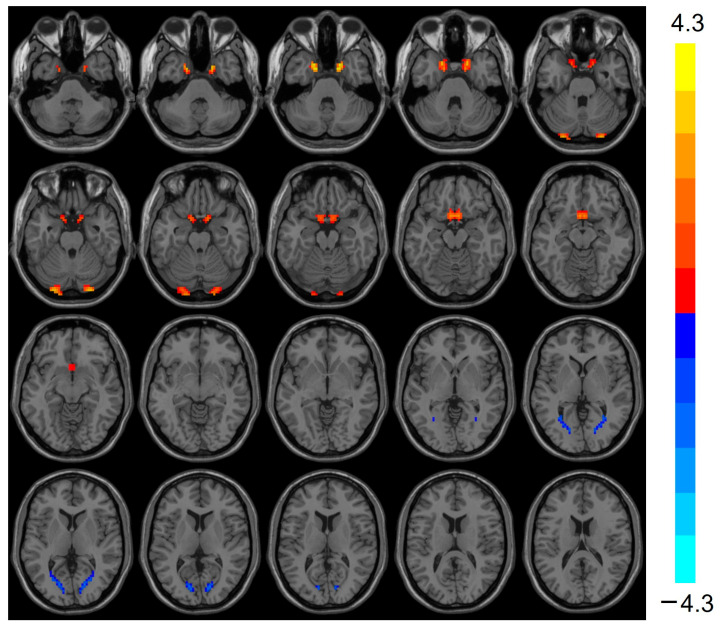
VMHC values were elevated in the heart transplantation group in bilateral Cerebelum_Crus1 (red), whereas they were reduced in bilateral calcarine (blue). Color bars are t-values. VMHC, voxel-mirrored homotopic connectivity.

**Figure 3 biomedicines-13-00873-f003:**
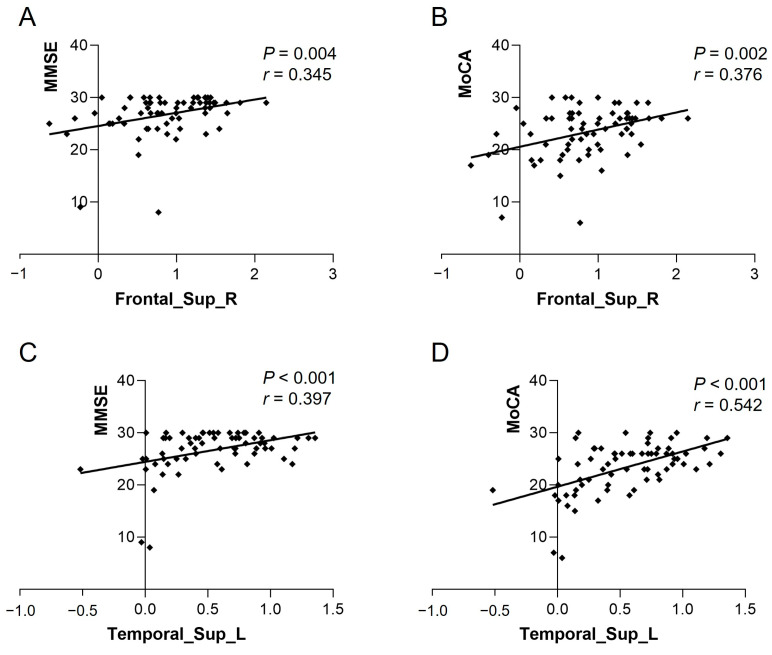
ReHo values of Frontal_Sup_R in the heart transplant group were positively correlated with (**A**) MMSE (*r* = 0.345, *p* = 0.004) and (**B**) MoCA (*r* = 0.376, *p* = 0.002). ReHo values of Temporal_Sup_L were positively correlated with (**C**) MMSE (*r* = 0.397, *p* < 0.001) and (**D**) MoCA (*r* = 0.542, *p* < 0.001). ReHo, regional homogeneity; MoCA, Montreal Cognitive Assessment; MMSE, Mini-Mental State Examination.

**Figure 4 biomedicines-13-00873-f004:**
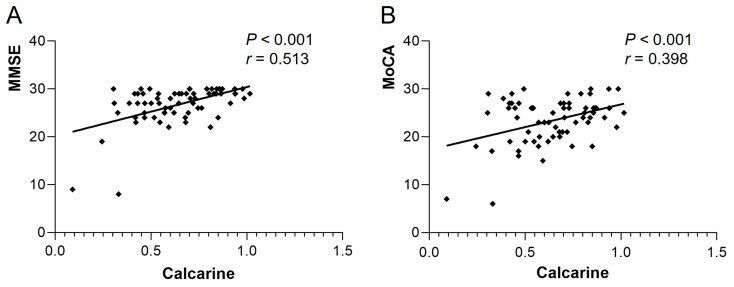
VMHC of bilateral calcarine in the heart transplant group showed a positive correlation with (**A**) MMSE (*r* = 0.513, *p* < 0.001) and (**B**) MoCA (*r* = 0.398, *p* < 0.001). VMHC, voxel-mirrored homotopic connectivity; MoCA, Montreal Cognitive Assessment; MMSE, Mini-Mental State Examination.

**Table 1 biomedicines-13-00873-t001:** Comparison of basic data between two groups.

Variable	HT	HC	t/χ^2^	*p*-Value	Cohen’s d	95% CI
Number of subjects	68	56	-	-	-	-
Age (years)	49.04 ± 12.57	51.77 ± 10.43	1.319	0.190	0.234	(−0.122, 0.588)
Gender (male/female)	42/26	26/30	2.916	0.088	-	-
Education (years)	11.90 ± 3.90	13.02 ± 3.68	1.635	0.105	0.295	(−0.061, 0.650)
MMSE score	26.72 ± 4.06	28.75 ± 0.792	4.028	<0.001	0.664	(0.299, 1.026)
MoCA score	23.38 ± 4.84	26.80 ± 2.81	4.914	<0.001	0.845	(0.474, 1.212)
Visuospatial/executive	3.13 ± 1.18	4.57 ± 0.57	7.175	<0.001	1.510	(1.109, 1.911)
Naming	2.52 ± 0.80	2.86 ± 0.36	2.519	0.014	0.531	(0.170, 0.891)
Attention	5.08 ± 1.13	5.82 ± 0.48	3.972	<0.001	0.825	(0.456, 1.193)
Language	1.65 ± 0.81	2.39 ± 0.69	4.090	<0.001	0.976	(0.602, 1.350)
Abstraction	1.19 ± 0.70	1.89 ± 0.31	5.988	<0.001	1.252	(0.865, 1.639)
Memory	0.63 ± 0.89	3.57 ± 1.45	9.729	<0.001	2.499	(2.027, 2.972)
Orientation	5.88 ± 0.33	5.93 ± 0.26	0.727	0.470	0.166	(−0.187, 0.520)

HT, heart transplantation; HC, healthy control; MoCA, Montreal Cognitive Assessment; MMSE, Mini-Mental Status Examination.

**Table 2 biomedicines-13-00873-t002:** Brain regions with differences in ReHo between groups.

Regions	Clusters Voxels	Peak MNI Coordinate			T-Values
		x	y	z	
Frontal_Sup_R	43	24	6	54	−4.422
Thalamus_L	29	−9	−30	−3	−3.911
Calcarine_L	34	−12	−84	6	−3.640
Temporal_Sup_L	26	−36	−21	9	4.609

ReHo, regional homogeneity; MNI, Montreal Neuroscience Institute.

**Table 3 biomedicines-13-00873-t003:** Brain regions with differences in VMHC between groups.

Regions	Clusters Voxels	Peak MNI Coordinate			T-Values
		x	y	z	
Cerebelum_Crus1_L	22	−27	−93	−24	3.803
Cerebelum_Crus1_R	22	27	−93	−24	3.803
Calcarine_L	49	−15	−78	9	−3.424
Calcarine_R	49	15	−78	9	−3.424

VMHC, voxel-mirrored homotopic connectivity; MNI, Montreal Neuroscience Institute.

## Data Availability

The data that support the findings of this study are available from the corresponding author.

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
