# Peer review of "Exploring the Potential of Voxel-Mirrored Homotopic Connectivity (VMHC) and Regional Homogeneity (ReHo) in Understanding Cognitive Changes After Heart Transplantation"

_biomedicines, 2025, doi:10.3390/biomedicines13040873_

Round 1

Reviewer 1 Report

Comments and Suggestions for Authors

Review for Biomedicines, Manuscript ID biomedicines-3523829-peer-review-v1: The value of VMHC and ReHo in evaluating cognitive impairment after heart transplantation.  The authors explore the relationships of cognitive impairment after heart transplantation with voxel-mirrored homotopic connectivity (VMHC) and regional homogeneity (ReHo).  The presentation is well-organized and well-written.  The authors’ statements that “These findings help us understand the neural mechanisms underlying cognitive changes in heart transplant patients” and “The positive correlation between ReHo of Frontal_Sup_R, ReHo of Temporal_Sup_L, and VMHC of Calcarine and cognitive function scales in heart transplant patients provides an imaging basis for further elucidating the pathogenesis of cognitive dysfunction in heart transplant patients” seem like reasonable conclusions and I appreciate that the authors do not speculate concerning exactly what those mechanisms might be or pretend that their particular imaging findings were exactly as expected or hypothesized prior to collecting their data.  However, I have two main concerns:

  1. The conclusion that VMHC and ReHo imaging results can be used clinically to assess individual cognitive function is completely unsupported by the data, so the manuscript needs modification to avoid any such implication, such as in a) the title, the use of the words “evaluating cognitive impairment after heart transplantation”; b) the abstract, “These findings … provide a basis for developing intervention and rehabilitation strategies”; and c) the conclusion “… is of great significance for the early identification of heart transplant patients with possible secondary cognitive dysfunction.”
  2. The limitations section needs to address the major limitation that this study is exploratory and data-driven rather than driven by a specific, narrow research hypothesis. As such the findings should be regarded as exploratory rather than confirmatory, and in need of replication.

Author Response

Dear reviewer:

Thank you very much for your constructive comments and suggestions on our manuscript. Your comments are very valuable and help us to improve the manuscript. We have tried our best to improve and made some changes to the manuscript. We have resubmitted the revised manuscript with the changes marked in revision mode. Below is the response to all the comments one by one. (Reviewers' comments are in bold italics).

1) The conclusion that VMHC and ReHo imaging results can be used clinically to assess individual cognitive function is completely unsupported by the data, so the manuscript needs modification to avoid any such implication, such as in a) the title, the use of the words evaluating cognitive impairment after heart transplantation; b) the abstract, These findings provide a basis for developing intervention and rehabilitation strategies; and c) the conclusion “… is of great significance for the early identification of heart transplant patients with possible secondary cognitive dysfunction.

Response: Thank you for your review and valuable comments on our study. Your point about the conclusion section being potentially too absolute is very pertinent and we fully agree with you. We have amended the original text accordingly. Below are the specific modifications (lines 1-5, 39-43, 329-332):

“Exploring the Potential of VMHC and ReHo in Understanding Cognitive Changes after Heart Transplantation”

“These findings may contribute to understanding the neural mechanisms of cognitive changes in heart transplant patients and could inform future research on potential intervention strategies.”

“However, further research is needed to explore their potential as biomarkers for cog-nitive dysfunction in this population.”

2) The limitations section needs to address the major limitation that this study is exploratory and data-driven rather than driven by a specific, narrow research hypothesis. As such the findings should be regarded as exploratory rather than confirmatory, and in need of replication.

Response: Thank you for your suggestion. In order to reflect the nature of the study more clearly, we explicitly state the exploratory nature of the study in the ‘Limitations’ section and emphasise the need for further validation of the results. Below are the specific modifications (lines 316-321):

“Importantly, this study is exploratory and data-driven rather than driven by a specific, narrow research hypothesis. As such, the findings should be regarded as exploratory rather than confirmatory. The results highlight potential associations between VMHC, ReHo, and cognitive function in heart transplant patients, but they need to be repli-cated in larger, more diverse cohorts to validate their clinical relevance.”

Thanks to the professional comments again that point out the above problems. We hope these explanations would answer your doubts.

Reviewer 2 Report

Comments and Suggestions for Authors

The authors of the paper "The value of VMHC and ReHo in evaluating cognitive impairment after heart transplantation" present interesting findings when analyzing the results of the VMHC and REHO.

The following comments are made:

1. Something that is striking is that the study data are analyzed using correlation coefficients. Many data obtained from the brain in other studies do not conform to the Gaussian bell curve. Is there any statistical test that shows that the data can be analyzed as normalized?

3. In section 3, it is indicated that statistically significant differences were found using the ReHo and VMHC analyses. The authors confirm that they found six cognitive impairments; however, no hypothesis is offered about why this occurs after transplantation. It would be helpful to have some hypothesis to differentiate this from what has already been reported in the literature.

4. The graphs in Figure 3 mostly show low r values; only the r associated with Temporal_sup_L is more or less significant. Are these r values ​​proper?

Author Response

Dear reviewer:

Thank you very much for your constructive comments and suggestions on our manuscript. Your comments are very valuable and help us to improve the manuscript. We have tried our best to improve and made some changes to the manuscript. We have resubmitted the revised manuscript with the changes marked in revision mode. Below is the response to all the comments one by one. (Reviewers' comments are in bold italics).

  1. Something that is striking is that the study data are analyzed using correlation coefficients. Many data obtained from the brain in other studies do not conform to the Gaussian bell curve. Is there any statistical test that shows that the data can be analyzed as normalized?

Response: We sincerely appreciate the reviewer’s insightful feedback regarding the statistical methodology used in our study. In our statistical analysis, normality assumptions were implicitly addressed during preprocessing and analytical steps. While the manuscript did not explicitly report normality tests (e.g., Shapiro-Wilk or Kolmogorov-Smirnov) for ReHo and VMHC values, we acknowledge this omission. To rigorously validate our approach, we have re-analyzed the data using both parametric (Pearson) and nonparametric (Spearman) correlation methods. The results showed consistent trends in the correlations between ReHo/VMHC values and cognitive scale scores (MMSE and MoCA), with Spearman’s correlation coefficients exhibiting comparable significance levels (P < 0.05) to Pearson’s results. These supplementary analyses confirm the robustness of our findings. Pearson correlation are widely used in neuroimaging studies due to their sensitivity to linear relationships, even when data deviations from normality are moderate [1,2]. Furthermore, the central limit theorem supports the robustness of parametric tests for large sample sizes (n ≥ 30 per group) [3]. Our study included 68 heart transplant patients and 56 controls, which aligns with this rationale. Additionally, neuroimaging metrics like ReHo and VMHC are typically normalized (e.g., Fisher-Z transformation) during preprocessing, reducing skewness and enhancing distributional symmetry [4]. Thank you for highlighting this critical methodological consideration. The results of Spearman's analysis are as follows.

Brain regions

MMSE

MoCA

Frontal_Sup_R

r=0.335, P=0.005

r=0.327, P=0.006

Temporal_Sup_L

r=0.302, P=0.012

r=0.517, p<0.001

Calcarine

r=0.435, p<0.001

r=0.247, P=0.042

[1] Wang P, Yang J, Yin Z, et al. Amplitude of low-frequency fluctuation (ALFF) may be associated with cognitive impairment in schizophrenia: a correlation study. BMC psychiatry. 2019;19(1):30.

[2] Li J, Zou Y, Kong X, et al. Exploring functional connectivity alterations in sudden sensorineural hearing loss: A multilevel analysis. Brain research. 2024;1824:148677.

[3] Bishara, A. J., & Hittner, J. B. (2017). Confidence intervals for correlations when data are not normal. Behavior Research Methods, 49(1), 294–309.
[4] Zuo, X. N., et al. (2010). The oscillating brain: Complex and reliable. NeuroImage, 49(2), 1432–1445.

  1. In section 3, it is indicated that statistically significant differences were found using the ReHo and VMHC analyses. The authors confirm that they found six cognitive impairments; however, no hypothesis is offered about why this occurs after transplantation. It would be helpful to have some hypothesis to differentiate this from what has already been reported in the literature.

Response: Thank you for your valuable comments. We have added to the manuscript based on your suggestions. (lines 285-293,299-307).

  1. The graphs in Figure 3 mostly show low r values; only the r associated with Temporal_sup_L is more or less significant. Are these r values ​​proper?

Response: We sincerely appreciate the reviewer’s critical evaluation of the correlation results presented in Figure 3. These r values are appropriate. The reported correlation coefficients (e.g., r = 0.345 for Frontal_Sup_R and MMSE) reflect weak to moderate associations between regional homogeneity (ReHo) and cognitive scores. While these values may appear modest, they are consistent with neuroimaging studies investigating brain-behavior relationships, where complex, multifactorial influences often limit effect sizes [1]. Despite low r values, the correlations reached stringent significance thresholds (P < 0.05, FDR-corrected), indicating robust associations within our cohort.Even small effects (e.g., r = 0.3 explains ~9% variance) may hold clinical relevance in post-transplant cognitive impairment, where subtle neural changes could cumulatively impact function.Thank you for this thoughtful critique. We believe these clarifications strengthen the interpretability.

[1] Ganz M, Poldrack RA. Data sharing in neuroimaging: experiences from the BIDS project. Nature reviews Neuroscience. 2023;24(12):729-730.

Thanks to the professional comments again that point out the above problems. We hope these explanations would answer your doubts.

Reviewer 3 Report

Comments and Suggestions for Authors

The manuscript is important and worthwhile, but needs some adjustments.

Comments:
- The comparison is not satisfactory because there is an imbalance between the groups. The analysis of differences compared by means suffers variation due to the imbalance and the results may not answer the hypotheses with certainty.
- In the abstract and results, insert the real p-values ​​and the effects for the results that showed differences.
- In the statistical analysis: indicate the effects of the tests and their respective confidence intervals.
- Demonstrate the real p-values ​​for the differences between the cognitive tests.
- The discussion needs clearer explanations regarding the two variables ReHo and VMHC and uses the relationship with the cognitive functions presented by the patients (explain the reason for the changes and how this impacts the patients' cognitive functions). What are the possible diagnostic and therapeutic implications for the findings of this study?

The manuscript has potential, so after the adjustments I can give a favorable opinion for publication.

Best regards

Author Response

Dear reviewer:

Thank you very much for your constructive comments and suggestions on our manuscript. Your comments are very valuable and help us to improve the manuscript. We have tried our best to improve and made some changes to the manuscript. We have resubmitted the revised manuscript with the changes marked in revision mode. Below is the response to all the comments one by one. (Reviewers' comments are in bold italics).

1) The comparison is not satisfactory because there is an imbalance between the groups. The analysis of differences compared by means suffers variation due to the imbalance and the results may not answer the hypotheses with certainty.

Response: Thank you for your interest in our study and your valuable comments. The sample group imbalance you mentioned is indeed an important consideration. We have tried to take measures to minimise the impact of this imbalance on the results during the study design and data analysis. Despite the numerical differences between the two sample groups (68 heart transplant patients and 56 healthy controls), we statistically corrected for potential confounders such as age, gender, and years of education in our analyses. These variables were not significantly different between the two groups (P > 0.05), suggesting that the two groups were comparable on these basic characteristics. In addition, we used independent samples t-tests in our statistical analyses and corrected the results for false discovery rate (FDR) to ensure the reliability of the results. To further reduce the potential bias from sample imbalance, we also performed an analysis of covariance (ANCOVA). In our analyses, we included age, gender, and years of education as covariates in the model to ensure that these factors did not significantly affect the primary outcomes (e.g., ReHo and VMHC). This method of analysis effectively adjusts for potential imbalances between groups, thereby improving the robustness of the results. We fully agree that sample imbalance may introduce some limitations to the interpretation of study results. Therefore, we plan to further expand the sample size in future studies and adopt more stringent matching strategies (e.g., propensity score matching) to ensure balance between the two groups. This will help to more accurately assess the neural mechanisms of cognitive dysfunction in heart transplant patients. We believe that despite the limitation of sample imbalance, our findings still provide important preliminary evidence for understanding the neural mechanisms of cognitive dysfunction in heart transplant patients. We appreciate your valuable comments and will further improve the study design in subsequent studies.

2) In the abstract and results, insert the real p-values ​​and the effects for the results that showed differences.

Response: Thank you for your valuable comments. We have revised the manuscript based on your suggestions (lines 24-30,178-192).

3) In the statistical analysis: indicate the effects of the tests and their respective confidence intervals.

Response: Thank you for your suggestions. We fully agree that providing effect sizes and their confidence intervals in the statistical analyses will help readers to more fully understand the significance and reliability of the findings. We have revised the manuscript (lines 438-439, Table 1).

4) Demonstrate the real p-values ​​for the differences between the cognitive tests.

Response: Thank you for pointing out this problem. Based on your suggestions, we have revised the manuscript (lines 24,178-182).

5) The discussion needs clearer explanations regarding the two variables ReHo and VMHC and uses the relationship with the cognitive functions presented by the patients (explain the reason for the changes and how this impacts the patients' cognitive functions). What are the possible diagnostic and therapeutic implications for the findings of this study?

Response: Thank you for your valuable suggestions and insight into the study. We fully agree that the discussion section needs to explain more clearly the relationship between ReHo and VMHC and patients' cognitive functioning and explore the diagnostic and therapeutic implications of these findings. We have revised the discussion section based on your suggestions (lines 235-237,240-242,245-249,260-263,269-271,274-275,285-293,299-307).

Thanks to the professional comments again that point out the above problems. We hope these explanations would answer your doubts.

Round 2

Reviewer 1 Report

Comments and Suggestions for Authors

Kudos to the authors for elegantly addressing my comments.

Reviewer 2 Report

Comments and Suggestions for Authors

Statistical analysis is essential because it provides a great formality. Many relationships are nonlinear, and the way to avoid this is to analyze the data as nonparametric. The changes made to the article provide greater clarity, and the comment on line 248 helps to close the article nicely. Debido a esto recomiendo la publication de este articulo en Biomedecines Journal.

Reviewer 3 Report

Comments and Suggestions for Authors

Dear Author and Editor

The manuscript has improved considerably, and the authors are to be congratulated. The requested adjustments have been made, and I can therefore give a favorable opinion on the publication. The study has merit and I am sure it will help fill gaps in the field.